# Right Ventricular Diastolic Dysfunction after Marathon Run

**DOI:** 10.3390/ijerph17155336

**Published:** 2020-07-24

**Authors:** Zuzanna Lewicka-Potocka, Alicja Dąbrowska-Kugacka, Ewa Lewicka, Rafał Gałąska, Ludmiła Daniłowicz-Szymanowicz, Anna Faran, Izabela Nabiałek-Trojanowska, Marcin Kubik, Anna Maria Kaleta-Duss, Grzegorz Raczak

**Affiliations:** 1Department of Cardiology and Electrotherapy, Medical University of Gdańsk, 80-210 Gdańsk, Poland; zuzanna.lewicka@gumed.edu.pl (Z.L.-P.); ewa.lewicka@gumed.edu.pl (E.L.); ludmila.danilowicz-szymanowicz@gumed.edu.pl (L.D.-S.); anfar@wp.pl (A.F.); izabela.nabialek-trojanowska@gumed.edu.pl (I.N.-T.); mkubik@gumed.edu.pl (M.K.); ania.m.kaleta@gmail.com (A.M.K.-D.); grzegorz.raczak@gumed.edu.pl (G.R.); 2First Department of Cardiology, Medical University of Gdańsk, 80-210 Gdańsk, Poland; rgal@gumed.edu.pl

**Keywords:** marathon run, amateur runners, diastolic function, right ventricle, relaxation, isovolumic relaxation time, myocardial performance index

## Abstract

It has been raised that marathon running may significantly impair cardiac performance. However, the post-race diastolic function has not been extensively analyzed. We aimed to assess whether the marathon run causes impairment of the cardiac diastole, which ventricle is mostly affected and whether the septal (IVS) function is altered. The study included 34 male amateur runners, in whom echocardiography was performed two weeks before, at the finish line and two weeks after the marathon. Biventricular diastolic function was assessed not only with conventional Doppler indices but also using the heart rate-adjusted isovolumetric relaxation time (IVRTc). After the run, IVRTc elongated dramatically at the right ventricular (RV) free wall, to a lesser extent at the IVS and remained unchanged at the left ventricular lateral wall. The post-run IVRTc_IVS correlated with IVRTc_RV (r = 0.38, *p* < 0.05), and IVRTc_RV was longer in subjects with IVS hypertrophy (88 vs. 51 ms; *p* < 0.05). Participants with measurable IVRT_RV at baseline (38% of runners) had longer post-race IVRTc_IVS (102 vs. 83 ms; *p* < 0.05). Marathon running influenced predominantly the RV diastolic function, and subjects with measurable IVRT_RV at baseline or those with IVS hypertrophy can experience greater post-race diastolic fatigue.

## 1. Introduction

There is increasing evidence that prolonged intense exercise, such as marathon running, can affect cardiac performance [1]. Recently, this form of sports activity has gained popularity, but the question remains about its safety for amateur non-elite runners, who are often middle-aged [2].

It has been raised that the right ventricle (RV) may be the “Achilles heel” of the competing heart, because RV enlargement and reduction in RV contractility was observed following marathon running [3]. An increase in oxygen demand during prolonged exercise, followed by the augmentation of cardiac output and a proportionate rise in pulmonary artery pressure, ultimately affects RV, which is not adapted to high vascular resistance [4,5]. In the presence of common septum and pericardium constraint, dysfunction of one ventricle affects the other in the process of ventricular interdependence. Apart from their dominant role in this interaction, the septal muscle fibers that obliquely bind both ventricles also fulfil a major function in the overall movement of the RV [6,7]. Exercise-induced cardiac fatigue has been studied mainly in terms of depressed systolic function [8]. In contrast, a small number of studies have assessed the cardiac diastole among amateur marathon runners from pre-event to in-run testing, especially regarding RV. According to the theory of myocardial damage progression, the diastole tends to become impaired before the systole, and, in several diseases, the slowing of relaxation was shown as an early sign of myocardial damage [9,10,11]. The prolongation of the isovolumic relaxation time (IVRT) may reflect the delayed relaxation filling pattern and therefore be the first marker of developing diastolic dysfunction [12]. In this study, we analyzed the pathophysiology of heart exhaustion associated with a marathon competition, with special attention to diastolic function and interventricular septum involvement.

## 2. Materials and Methods

After having received acceptance of the study protocol from the Independent Bioethics Commission for Research of the Medical University of Gdansk (NKBBN/104/2016), via advertisements sent to local sport clubs, we recruited male amateur marathon runners planning to run in the 2nd PZU Marathon in Gdańsk, Poland. The study was conducted in accordance with the Declaration of Helsinki. Detailed study information was provided to all volunteers and written consent was obtained from each participant before entering the study. We enrolled subjects who were at defined age (20–55 years old) and showed no chronic diseases. The study protocol consisted of three stages and participants were examined two weeks before the marathon run (stage I), at the marathon finish line (stage II) and two-weeks after the competition (stage III). At each stage, physical and echocardiographic examination (ECHO) was performed. Moreover, at baseline (stage I), the training history was collected and a cardiopulmonary exercise test (CPET) was performed. Detailed characteristics of the examined amateur marathon runners have been previously presented [13]. During the competition, participants were allowed to rehydrate on a whim and no food intake restrictions were advised. The temperature was around 12 °C and the wind speed was around 21.5 km/h.

The ECHO was carried out with a commercially available system (Vivid E9, GE Healthcare, Horten, Norway), in accordance with the recommendations of the American Society of Echocardiography and European Association of Cardiovascular Imaging [14]. All analyses and measurements (averaged from three consecutive beats) were performed off-line by two researchers, using echocardiographic quantification software (EchoPac 201, GE Healthcare, Norway). By means of two-dimensional (2D) and M-mode ECHO, cardiac dimensions were obtained, including diastolic interventricular septum diameter (IVSd) in the parasternal long-axis view (PLAX). The end-systolic right atrial (RA) area was calculated in the apical four-chamber (A4C) view and the left atrial (LA) volume indexed to body surface area (LAVI) was obtained from the apical two-chamber and A4C views. At end-diastole, the basal LV and RV transversal dimensions (LVd BAS and RVd BAS) were acquired in the A4C view and, subsequently, the RVd/LVd BAS ratio was calculated to assess diastolic ventricular interaction. The transverse RV diameter was also measured in the middle of the RV inflow (RVd MID) in the RV-focused apical view, as described in detail previously [14].

The diastolic function of LV and RV was assessed with pulsed wave Doppler (PWD) and spectral Doppler tissue imaging (DTI) indices according to the guidelines [15,16]. In the 4C view, the PWD transmitral (MV) and transtricuspid (TV) flow velocities were obtained: peak early (E), peak atrial (A) and E/A ratio. Spectral DTI mitral annular velocities, namely peak systolic (S’), peak early diastolic (E’), peak atrial diastolic (A’), E’/A’ ratio, were assessed at basal septum (S’IVS, E’IVS, A’IVS) and LV lateral wall (S’LW, E’LW, A’LW). Then, the following ratios were calculated: E_MV/E’_LW, E_MV/E’_IVS and E_MV/E’_AVG (E’ averaged from IVS and LW) [16]. Corresponding tricuspid annular velocities were obtained (S’RV, E’RV, A’RV), along with E_TV/E’_RV ratio. Spectral DTI isovolumic relaxation time (IVRT) was estimated at the lateral mitral (IVRT_LW), tricuspid RV (IVRT_RV) and septal (IVRT_IVS) level, as the interval between the end of the S’-wave and the beginning of the E’-wave, as described previously [17,18]. In order to compensate for the augmented heart rate after exercise, the heart rate-adjusted IVRT (IVRTc) was calculated as the ratio of the IVRT and square RR (interval between two subsequent beats, given in seconds).

RV and LV global performance was assessed by the DTI-derived myocardial performance index (MPI), which was analyzed in the 4C view at the lateral tricuspid (MPI_RV), lateral mitral (MPI_LW) and septal annulus (MPI_IVS). MPI was calculated with the formula (ICT + IVRT)/ET, where ICT is the isovolumic contraction time, measured from the cessation of A′ -wave to the onset of S’-wave, and ET is the ejection time, measured as the width of S’-wave. The DTI-derived MPI was chosen over the PWD as it is derived from one cardiac cycle and therefore is more reliable [19].

The RV four-chamber longitudinal strain (including ventricular septum), here also referred to as “RV global strain” (RV 4CSL) was measured in the RV-focused 4C view, in accordance with the consensus document on deformation imaging [20]. Along with the guidelines, tricuspid annular plane systolic excursion (TAPSE) and RV fractional area change (RV FAC) were additionally calculated to measure the RV systolic function, and regarding the LV ejection fraction (LV EF) and LV global longitudinal strain (LV GLS), assessment was performed [14,15].

The study subjects were divided into 2 groups on the basis of the baseline IVRT_RV: (1) IVRT_RV = 0 ms or (2) IVRT_RV > 0 ms. Another division was performed according to the presence of IVS hypertrophy, defined as IVSd > 11 mm.

Data analysis was performed using Statistica 13.3 software (Statsoft Inc., Tulsa, Oklahoma, United States). The normality of variables was tested with the Shapiro–Wilk test. Data are presented as mean ± SD (if normally distributed) or median with first and third quartile (25th; 75th percentile) (if non-normally distributed). The comparison between 3 stages was performed with ANOVA analysis and the post-hoc Tukey test for normally distributed data. Non-normally distributed measurements were compared with Friedman ANOVA and post-hoc for Friedman ANOVA. Comparisons between predefined groups were performed by Student’s t-test for independent samples or the Mann–Whitney U test, where appropriate. The *p*-value of < 0.05 was considered significant. Spearman’s correlation was calculated to determine the dependency of ECHO measurements and parameters of cardiorespiratory fitness obtained in the CPET.

## 3. Results

Thirty-four amateur marathon runners, with a mean age of 40 ± 8 years, who finished the competition, were enrolled in the study. All participants were men of Caucasian race with no relevant medical history. Training history and CPET results have been recently published [13]. In brief, the mean training distance was 56.5 ± 19.7 km/week, and the mean marathon finishing time was 3.7 ± 0.4 h. The mean oxygen uptake at anaerobic threshold (VO_2_AT) was 39.7 ± 6.9 mL/kg/min. Table 1, Table 2 and Table 3 provide the comparison of 2D ECHO measurements between three stages, including measures of both ventricles (Table 1), Doppler parameters of LV performance (Table 2) and the RV Doppler indices (Table 3).

There were no significant differences between parameters from stages I and III, apart from E_MV and E’_RV (Table 2 and Table 3). Running the marathon had no impact on the LV systolic function (Table 1). There were no relevant valvular regurgitations among marathon runners at any stage of the study. Reduced ratios of E/A_MV and E/A_TV were observed after the competition (Table 2 and Table 3). The results obtained by PWD were consistent with DTI-derived E’/A’ in both RV and LV (Table 2 and Table 3). On the contrary, the E/E’ ratios remained at the same level after the race (Table 2 and Table 3). The most striking evidence of impaired RV relaxation was shown on the basis of the IVRT analysis (Figure 1).

The marathon run resulted in significant prolongation of IVRTc_RV, which was found in 24 out of 34 runners (71%). The IVRT_RV appeared in 58% of subjects, in whom it was undetectable at the baseline. The marathon impact on the relaxation was different for the RV and LV: immediately after the run, IVRTc elongated dramatically at the RV free wall, to a lesser extent at the IVS and remained at the same level at the LV lateral wall (Figure 2).

In post-marathon analysis, there was a correlation between IVRTc_RV and IVRTc_IVS (r = 0.38, *p* < 0.05), and IVRTc_RV was significantly longer in runners with IVS hypertrophy (88 vs. 51 ms; *p* < 0.05). At the baseline, 13 (38%) participants presented measurable IVRT_RV, and in this group, a longer post-race IVRTc_IVS was revealed (102 vs. 83 ms; *p* < 0.05).

MPI measurements paralleled the IVRT results. After the race, a significant prolongation at the RV free wall was observed, accompanied by a trend towards its increase at IVS (*p* = 0.09) but unchanged MPI at the LV lateral wall (Table 2 and Table 3). There was a post-marathon reduction in RV deformation, but other 2D parameters assessing RV systolic function did not change after the race (Table 1). The cardiorespiratory fitness assessed in CPET was a predictor of RV performance during marathon running. Participants with higher VO_2_AT had lower MPI_RV post-race (r = −0.41, *p* < 0.05).

After the race, the RV enlargement and diminishment of the LV diameter was observed: RVd MID and RVd/LVd BAS became significantly larger and LVd BAS significantly decreased (Table 1). As presented previously, the median IVSd was 11 mm, with a range of 7–17 mm [13], and in nine participants who were significantly older (44 ± 4 vs. 37 ± 9 years, *p* < 0.05), IVSd was > 11 mm.

There were no differences in the pre- and post-marathon left and right atrial sizes, expressed as LAVI and RA area. Nevertheless, there was a correlation between the post-marathon RA enlargement and IVRT_RV (r = 0.48, *p* < 0.05) and MPI_RV (r = 0.58, *p* < 0.05). Moreover, in the post-race analysis, larger RA areas were found in subjects with more evident diastolic ventricular interaction, as indicated by an increased RVd/LVd BAS ratio (r = 0.49, *p* < 0.05). Notably, participants with IVRT_RV > 0 ms at baseline and those with IVS hypertrophy (IVSd > 11 mm) presented significantly larger LAVI and RA areas after the run (*p* < 0.05).

## 4. Discussion

There is an ongoing debate regarding whether intense exercise, such as running a marathon, is harmful to an overloaded heart [1]. Even if marathon-induced changes in cardiac function develop, they are difficult to register and show, as documented by the discrepancies between reports on this topic. In this study, we demonstrated the post-run RV enlargement, the increase in the RV/LV ratio and ambiguous findings on the RV systolic function (manifested in the decline in RV global strain but not in TAPSE, FAC or S’RV). The analysis of MPI, which reflects both systolic and diastolic function, showed changes only at the RV free wall. Our study revealed RV diastolic dysfunction manifesting as the post-run prolonged relaxation, especially at the RV free wall and to a lesser degree at the IVS. Although we found differences in the post-race E/A ratios for both ventricles, we do not interpret these results as diastolic impairment but rather as the marathon-induced impact of increased heart rate. All abnormalities were transient and not observed in the control examination performed two weeks after the marathon run. DTI-derived IVRTc proved to be the most sensitive marker of the RV diastolic failure among amateur marathon runners. We suggest that the threshold level of exercise that causes myocardial damage is different for RV and LV, as there was no change in IVRTc at the LV lateral wall.

With the variety of possible diastolic parameters, a question arises which should be considered in amateur marathon runners. The diastole consists of isovolumic relaxation, rapid filling, diastasis and atrial contraction, and each can be assessed depending on clinical indications [12]. With reference to RV, the guidelines suggest the assessment of transtricuspid E/A ratio, E/E’ ratio and RA size [15]. The recognition of abnormal LV diastolic function in patients with preserved LV EF relies on E’ velocities, average E/E’ ratio, LAVI and peak tricuspid regurgitation velocity [16]. Nevertheless, due to the preload and afterload-dependence of the diastolic indices, their measurements should be interpreted with caution [16].

With reference to diastolic function, the majority of previous studies among marathon runners have focused mainly on the LV and documented generally the post-race decrease in E/A ratio [21,22,23,24]. In our study we revealed similar findings for both ventricles, with mitral and transtricuspid inflow velocities having acted similarly. However, Doppler measurements have some limitations as they are strongly load-dependent [15,25]. Thus, the hydration status of runners is relevant, as fluid loss causes the decrease in E-, A-velocities and E/A ratio [26]. Moreover, as the duration of diastole is inversely correlated with heart rate, the atrial contraction gains significance over the rapid filling in case of tachycardia, resulting in decreased E/A ratio [15,27]. Undoubtedly, such results do not necessarily reflect diastolic dysfunction, as the E/E’ ratio after the run remained at the same level. Therefore, we argue for the utility of PWD indices in recognition of diminished ventricular relaxation among marathon runners.

IVRT is a more reliable parameter and among Doppler indices is the earliest to alter in the case of impaired relaxation [12]. Furthermore, it is reproducible and easy to obtain. It has been shown that inadequate IVRT_LV shortening during dobutamine stress echocardiography was the only Doppler diastolic parameter able to discriminate patients with residual ischemia after myocardial infarction [18]. With reference to RV, IVRT_RV correlates well with the RV systolic pressure and this non-invasive parameter is able to distinguish patients with elevated pulmonary pressure from those without [17]. Additionally, invasively measured early RV relaxation is abnormal in patients with pulmonary hypertension with preserved RV contractility [11].

In previous reports on marathon runners, no significant post-race alterations were noticed in IVRT_LV, which is consistent with our results [28,29]. However, in this study, IVRT was measured additionally for the RV free wall and septum, and to preclude the impact of heart rate increase during exercise, the heart rate-adjusted IVRT (IVRTc) was calculated. To our knowledge, our study is the first to report on the pre- and post-marathon IVRTc within the LV and RV. We showed the acute post-marathon impairment of the RV relaxation, manifesting by IVRTc_RV prolongation.

In contrast to the LV, in a well-functioning RV, we rather expect IVRT to be undetectable. IVRT becomes measurable when RV impairment occurs or when RV has to overcome a significant increase in pulmonary artery pressure [19]. The fact that more than half of runners who had IVRT_RV undetectable at baseline developed it after the race is alarming and indicates that marathon running can seriously overload the RV and alter its performance. The prolongation of RV relaxation probably reflects the exercise-induced augmentation of pulmonary artery pressure. No increment in tricuspid or pulmonary regurgitation was registered in ECHO performed several minutes after the race; therefore, the pressure increase seems transient. Exercise ECHO studies in patients with pulmonary hypertension do confirm that the increase in peak tricuspid regurgitation velocity returns to baseline values briefly after exercise cessation [30]. The analysis of IVRT_RV allows us to monitor changes in pulmonary vasculature, independently of tricuspid regurgitation velocity.

The recognition of detectable IVRT_RV at the baseline in 38% of the studied runners suggests the existence of pre-marathon RV overload or even subclinical damage in these subjects. Most strikingly, they showed more evident post-race diastolic impairment, with a significant increase in atrial sizes and longer IVS relaxation. The presence of measurable resting IVRT_RV raises questions about the individual’s upper limit of endurance exercise and their predisposition to training-induced cardiac fatigue. We hypothesize about the possibility of a “cumulative” alteration of the RV diastole due to repetitive RV exhaustion. Previously, it was demonstrated that greater marathon-induced decrement of RV contractility happened in less-trained runners [24]. Though we did not find a direct correlation between changes in IVRT_RV and the amount of training, cardiorespiratory fitness assessed in CPET was a predictor of global RV performance during the marathon run.

The influence of the marathon run on the LV relaxation was complex. There was no significant change in IVRTc_LW, but interestingly, the IVRTc_IVS was extended. Therefore, the impairment of septal relaxation mirrored abnormalities in the RV free wall. Changes in septal function may be explained as alterations in continuity with RV damage. Although IVS is mainly a constituent part of the LV, it also shares muscle fibers with RV [31]. Therefore, IVS is involved in the functioning of both RV and LV and also transmits the altered load conditions from one ventricle to another [6]. Our study demonstrates the post-marathon increase in diastolic ventricular interaction with enlarged RV and reduced LV. In accordance with former research that showed the post-run right to left IVS displacement [32], we speculate that another factor that could account for impaired septal relaxation is enhanced ventricular dependence.

Due to its complicated structure, an accurate 2D analysis of the RV’s performance remains challenging and often requires multiparametric assessment [33]. We revealed the post-marathon decline in global RV function, demonstrated by MPI, which combines systolic and diastolic function and does not depend on heart rate or ventricular geometry [34]. In previous studies, MPI was shown to correlate well with RV EF derived from cardiac magnetic resonance (CMR) imaging and was used to assess ventricular function in many diseases [34,35,36]. In pulmonary arterial hypertension, MPI_RV is successful in determining the severity of the disease and in patient monitoring [36]. To our knowledge, our study is the first reporting on the pre- and post-endurance MPI and for both LV and RV [37,38].

In contrast to RV, we found no significant change in MPI_LV, which confirms the notion that marathon running does not alter the LV diastole. Notably, only an insignificant trend of MPI_IVS prolongation was observed (*p* = 0.09). However, it is unclear whether this is due to the IVS involvement in RV diastolic impairment or whether it reflects a decrease in septal contractility. This requires further examination to determine whether septal diastolic dysfunction precedes its systolic impairment and whether a drop in IVS contractility appears at some point during repetitive marathon attendance. With reference to septum, the negative effect of “cumulative exercise dose” was previously proved in a CMR study that analyzed the late gadolinium enhancement and found that the occurrence of myocardial fibrosis in IVS near the RV attachment correlated with the longevity of sport competition [32]. In our study, we demonstrated that the effort-induced remodeling of IVS and the degree of its hypertrophy is relevant to the RV’s performance during the marathon run. Though IVS hypertrophy is generally considered a physiological adaptation to endurance exercise, we found that the presence of IVS with a width of > 11 mm appeared to be a predictor of more evident post-race diastolic fatigue.

Our results showed the impaired relaxation of the RV, with inconclusive observations regarding RV systolic function, as only RV deformation decreased. Additionally, we demonstrated that, after the marathon run, RV enlargement and LV diminishment appear. We presume that the sequence of cardiac changes started with elevated pulmonary pressure due to the exercise-induced increase in cardiac output, which influenced RV relaxation, as the first marker of developing dysfunction. Consequently, the RV enlarged and, due to pericardium constraint, the LV diminished its volume.

The observation regarding post-race RA size is also very interesting. Although we did not observe changes in atrial size in the whole group, post-race RA enlargement was greater in runners with longer post-race IVRT_RV, MPI_RV and larger RV volume. Additionally, in runners with impaired RV function at the baseline (IVRT_RV > 0 ms) and in those with hypertrophied septum, the atria after the run were bigger than in runners without these abnormalities.

Our study has an important limitation. As we did not include women in the study group, we cannot apply our findings to the population of female runners. Therefore, we are not able to discuss possible gender differences in RV diastolic fatigue caused by marathon running.

## 5. Conclusions

In amateur participants, the marathon run influences predominantly the RV diastole, and post-race IVRT assessment reveals the dramatic impairment of relaxation at the RV free wall, with concomitant alteration of the IVS. The division of LV_IVRTc into IVRTc_LW and IVRTc_IVS and their separate analyses is crucial, as marathon running does not influence the relaxation of the LV lateral wall. The RV MPI and RV strain analysis, showing the decline in RV global function, follows the IVRTc observations, demonstrating that marathon attendance overburdens mainly the RV. Nevertheless, RV overload through enhanced diastolic ventricular interaction also affects LV, causing a decrease in LV cavity and alteration of IVS function. At this point, we also indicate that there is a group of runners predisposed to occurrence of diastolic dysfunction after the marathon. Participants with evidence of detectable IVRT_RV at the baseline or those with IVS hypertrophy are endangered by greater post-race cardiac exhaustion. These subjects may require constant cardiac monitoring if they continue to exercise intensively.

## Figures and Tables

**Figure 1 ijerph-17-05336-f001:**
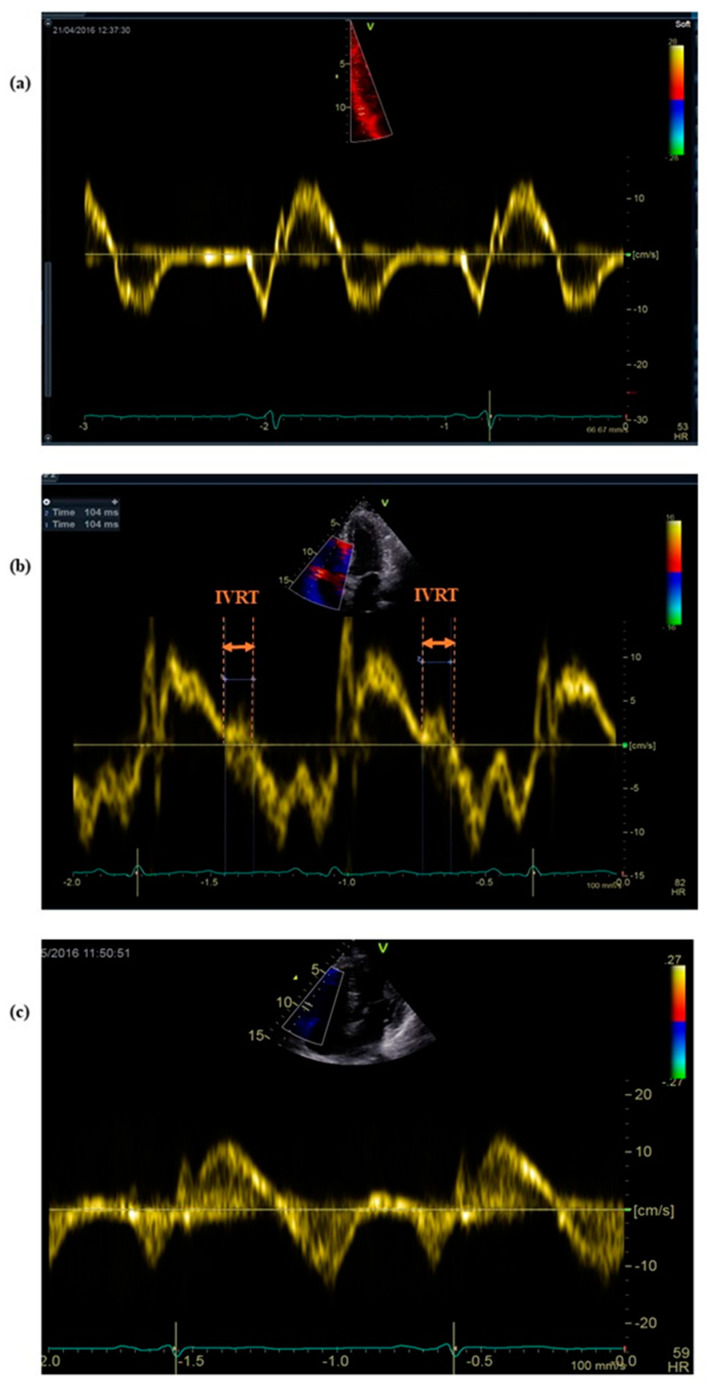
Changes in the right ventricular isovolumic relaxation time (IVRT_RV) between the three study stages. (**a**) IVRT_RV was undetectable at stage I (two weeks before the marathon run) and (**c**) at stage III (two weeks after the competition); (**b**) in contrast, the appearance of IVRT_RV and its prolongation up to 104 ms at stage II (at the marathon finishing line).

**Figure 2 ijerph-17-05336-f002:**
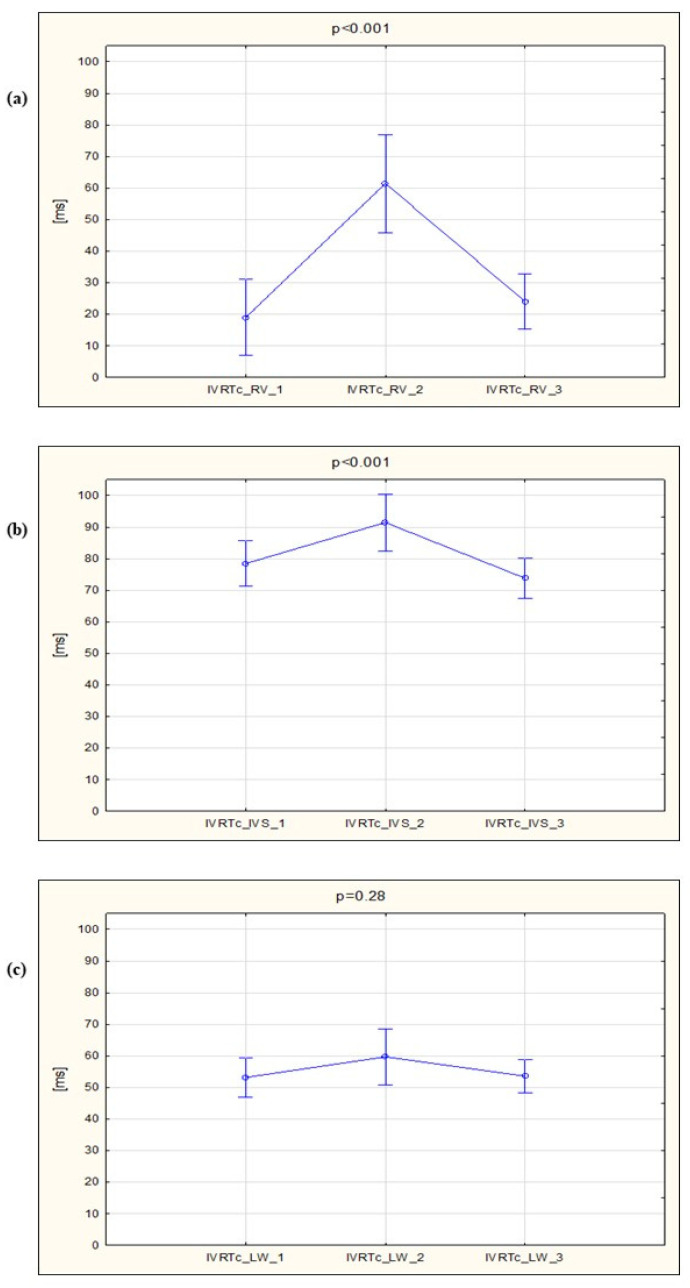
Changes in the heart rate-adjusted isovolumic relaxation time between the three stages of the study, assessed in the spectral Doppler tissue imaging (**a**) at the lateral tricuspid annulus (IVRTc_RV), (**b**) at septum (IVRTc_IVS) and (**c**) at the lateral mitral annulus (IVRTc_LW).

**Table 1 ijerph-17-05336-t001:** Echocardiographic parameters of the left and right ventricle obtained in amateur marathon runners.

Parameter	Stage I	Stage II	Stage III	ANOVA *p*-Value	Post-Hoc *p*-Value	
	Mean ± SD ^1^ or Median (1st; 3rd Quartile) ^2^				Stage I vs. II	Stage I vs. III
LV EF (%)	61.8 ± 4.9	60.5 ± 4.4	60.7 ± 4.5	* 0.38	-	-
LV GLS (%)	−19.9 ± 2.3	−19.4 ± 2.1	−19.7 ± 2.2	* 0.41	-	-
RV 4CSL (%)	−22.0 ± 2.8	−20.80 ± 2.6	−21.49 ± 2.5	* <0.05	<0.05	0.46
TAPSE (mm)	25.0 ± 3.6	24.0 ± 3.7	25.0 ± 2.7	* 0.56	-	-
RV FAC (%)	43 (37; 45)	39 (35; 44)	41 (36; 45)	^ 0.19	-	-
RVd MID (cm)	3.4 ± 0.6	3.7 ± 0.5	3.5 ± 0.5	* <0.01	<0.01	0.08
RVd BAS (cm)	3.8 ± 0.4	3.8 ± 0.5	3.9 ± 0.5	* 0.44	-	-
LVd BAS (cm)	4.8 ± 0.4	4.6 ± 0.3	4.9 ± 0.3	* <0.001	<0.01	0.88
RVd/LVd BAS	0.77 ± 0.1	0.82 ± 0.1	0.79 ± 0.1	* <0.05	<0.05	0.59

^1^—when normally distributed; ^2^—when non-normally distributed; Stage I—two weeks before the marathon run; Stage II—at the marathon finish line; Stage III—two weeks after the marathon run; LV —left ventricular; EF—ejection fraction; GLS—global longitudinal strain; RV—right ventricular; 4CSL—four-chamber longitudinal strain = global strain; TAPSE—tricuspid annular plane systolic excursion; FAC—fractional area change; RVd MID—RV mid-cavity end-diastolic dimension; LVd BAS—LV basal end-diastolic diameter; RVd BAS—RV basal end-diastolic diameter; RVd/LVd BAS—basal RV to LV end-diastolic diameter ratio; SD—standard deviation; * ANOVA with post-hoc Tukey test if applicable; ^ Friedman ANOVA with post-hoc average rank test if applicable.

**Table 2 ijerph-17-05336-t002:** Left ventricular parameters obtained by means of the pulsed wave Doppler and spectral Doppler tissue imaging in amateur marathon runners.

Parameter	Stage I	Stage II	Stage III	ANOVA *p*-Value	Post-Hoc *p*-Value	
	Mean ± SD ^1^ or Median (1st; 3rd Quartile) ^2^				Stage I vs. II	Stage I vs. III
S’_LW (cm/sec)	11 ± 3	11 ± 3	11 ± 3	* 0.88	-	-
E’_LW (cm/sec)	15 (12; 17)	12 (10; 15)	14 (13; 16)	^ <0.001	<0.05	ns
A’_LW (cm/sec)	8 (7;9)	10 (9; 11)	8 (7; 9)	^ <0.001	<0.05	ns
E’/A’_LW	1.8 (1.4; 2.1)	1.2 (1.0; 1.5)	1.9 (1.6; 2.3)	^ <0.001	<0.05	ns
IVRT_LW (ms)	53 ± 17	54 ± 19	56 ± 15	* 0.99	-	-
IVRTc_LW	53 ± 17	59 ± 23	54 ± 15	* 0.28	-	-
MPI_LW	0.41 ± 0.08	0.45 ± 0.17	0.42 ± 0.07	* 0.20	-	-
S’_IVS (cm/sec)	8 (8; 9)	9 (8; 10)	9 (7; 10)	^ 0.24	-	-
E’_IVS (cm/sec)	11 ± 2	10 ± 2	11 ± 2	* <0.001	<0.01	0.73
A’_IVS (cm/sec)	8 (8; 10)	10 (9; 11)	8 (7; 10)	^ <0.001	<0.05	ns
E’/A’_IVS	1.3 ± 0.4	1.0 ± 0.3	1.4 ± 0.4	* <0.001	<0.001	0.26
IVRT_IVS (ms)	82 (65; 95)	80 (68; 94)	78 (64; 86)	^ 0.46	-	-
IVRTc_IVS	78 (66; 92)	92 (77; 108)	73 (65; 85)	^ <0.001	<0.05	ns
MPI_IVS	0.55 (0.44; 0.59)	0.53 (0.44; 0.6)	0.47 (0.44; 0.54)	^ 0.09	-	-
E_MV (cm/sec)	71 (67; 87)	67 (55; 77)	78 (68; 92)	^ <0.01~	ns	ns
A_MV (cm/sec)	51 ± 10	65 ± 14	56 ± 11	* <0.001	<0.001	0.29
E/A_MV	1.5 ± 0.4	1.1 ± 0.3	1.5 ± 0.4	* <0.001	<0.001	0.7
E_MV/E’_LW	5.4 ± 1.2	5.5 ± 1.6	5.6 ± 1.8	* 0.66	-	-
E_MV/E’_IVS	7.1 ± 1.5	7.0 ± 1.8	7.6 ± 2.0	* 0.34	-	-
E_MV/E’_AVG	6.3 (5.2; 7.0)	5.8 (5.0; 7.2)	6.2 (5.4; 7.3)	^ 0.74	-	-

^1^—when normally distributed; ^2^—when non-normally distributed; LW—parameter measured at the lateral mitral annulus; IVS—parameter measured at the septal mitral annulus; S’—peak systolic tissue velocity; E’—peak early diastolic tissue velocity; A’—peak atrial diastolic tissue velocity; IVRT—isovolumic relaxation time; IVRTc—IVRT adjusted for heart rate; MPI—myocardial performance index; MV—mitral inflow; E—peak early flow velocity; A—peak atrial flow velocity; AVG—averaged for parameters obtained at IVS and LW; ns-*p*-value of >0.05 of post-hoc average rank test for Friedman ANOVA; ~ ANOVA test *p* <0.05, but post-hoc test revealed the difference between stages II and III, which was not the question of our study. * ANOVA with post-hoc Tukey test if applicable; ^ Friedman ANOVA with post-hoc average rank test if applicable. For other abbreviations, see Table 1.

**Table 3 ijerph-17-05336-t003:** Right ventricular parameters obtained by means of the pulsed wave Doppler and spectral Doppler tissue imaging in amateur marathon runners.

Parameter	Stage I	Stage II	Stage III	ANOVA *p*-Value	Post-Voc *p*-Value	
	Mean ± SD ^1^ or Median (1st; 3rd Quartile) ^2^				Stage I vs. II	Stage I vs. III
S’_RV (cm/sec)	14 (13; 16)	14 (13.5; 16)	15 (13; 16)	^ 0.51	-	-
E’_RV (cm/sec)	12 (11; 15)	12 (9; 14)	14 (13; 16)	^ <0.05 ~	ns	ns
A’_RV (cm/sec)	13 (10; 14)	16 (13; 20)	13 (12; 16)	^ <0.01	<0.05	ns
E’/A’_RV	1.0 (0.9; 1.2)	0.7 (0.6; 0.9)	1.2 (0.9; 1.3)	^ <0.001	<0.05	ns
IVRT_RV (ms)	0 (0; 29)	52 (32; 70)	21 (9; 34)	^ <0.001	<0.05	ns
IVRTc_RV (ms)	0 (0; 27)	58 (39; 78)	20 (0; 35)	^ <0.001	<0.05	ns
MPI_RV	0.28 (0.22; 0.37)	0.48 (0.35; 0.64)	0.33 (0.25; 0.41)	^ <0.001	<0.05	ns
E_TV (cm/sec)	55 ± 13	49 ± 11	56 ± 11	* 0.18	-	-
A_TV (cm/sec)	33 ± 10	46 ± 15	31 ± 8	* <0.001	<0.001	0.99
E/A_TV	1.7 ± 0.4	1.2 ± 0.3	1.9 ± 0.6	* <0.001	<0.001	0.55
E_TV/E’_RV	4.5 ± 1.1	4.2 ± 1.7	3.9 ± 1.0	* 0.44	-	-

^1^—when normally distributed; ^2^—when non-normally distributed; RV—right ventricular parameters measured at the lateral tricuspid annulus; TV—transtricuspid inflow. * ANOVA with post-hoc Tukey test if applicable; ^ Friedman ANOVA with post-hoc average rank test if applicable. For other abbreviations, see Table 1 and Table 2.

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
