# Peer review of "Right Ventricular Diastolic Dysfunction after Marathon Run"

_ijerph, 2020, doi:10.3390/ijerph17155336_

Round 1

Reviewer 1 Report

This article entitled "Right Ventricular Diastolic Dysfunction after Marathon Run'' is well designed and well written with scientific manners. The echocardiographic parameters seems acceptable to clarify the right ventricular dysfunction related after Marathon run properly. Thus, I have only a few minor points to comment. Although the authors mentioned about the relationship between volumetric change and RV dysfunction in line 182-192, please provide that whether RV and/or RA volumetric values were changed after marathon run which might affect the RV diastolic dysfunction and impaired relaxation as well as LV volumetric change. Nevertheless, I believe this study is valuable and could be extended to detect the high risk group for marathon run even in previous healthy adults.

Author Response

Dear Reviewer,

I would like to thank you for your comment on the manuscript ”Right ventricular diastolic dysfunction after marathon run”. Please, find the answers below.

Our results showed impaired relaxation of the RV, with inconclusive observations regarding RV systolic function, as only RV deformation decreased. Additionally, we demonstrated that after the marathon run RV enlargement and LV diminishment appeared. We presume that the sequence of cardiac changes started with elevated pulmonary pressure due to exercise-induced increase in cardiac output, which influenced RV relaxation, as the first marker of developing dysfunction. Consequently, the RV enlarged and due to pericardium constraint the LV diminished its volume.

The observation regarding post-race RA size is also very interesting. Although we did not observe changes in atrial size in the whole group, post-race RA enlargement was bigger in runners with longer post-race IVRT_RV, MPI_RV and larger RV volume. Additionally, in runners with impaired RV function at baseline (IVRT_RV > 0 ms) and in those with hypertrophied septum the atria after the run were bigger than in runners without these abnormalities (included in discussion lines 305-315).

All changes to the manuscript text are tracked in the revised manuscript file with the Track Changes' mode.

We hope that you will find our paper interesting and you will approve it for publication.

Yours faithfully,

Assoc. Prof. Alicja Dąbrowska-Kugacka, MD PhD

Zuzanna Lewicka, MD

Reviewer 2 Report

The manuscript ''Right Ventricular Diastolic Dysfunction after Marathon Run'' by Zuzanna Lewicka-Potocka et al. analyzes cardiac performance in the context of a marathon in male amateur runners. Herein, the authors report that the marathon run mainly affects the right ventricular diastolic function. Impaired relaxation at the right ventricular free wall occurred immediately after the race, and runners with IVS hypertrophy experienced higher diastolic fatigue. The study is well designed, and the data provided support the conclusions. I only have a few minor comments.

Minor comments:

  • Is there a specific reason for including only male runners in the study? The fact that men and women have different left ventricular dimensions and function is well-established. Furthermore, sex differences associate with cardiac adaptations to endurance exercise. Thus, failing to include both sexes could be a limitation of the study and should be addressed as part of the discussion.
  • Figure 2. The changes in the heart rate-adjusted isovolumic relaxation time (IVRTc_RV, IVRTc_IVS, and IVRTc_LW) between the three stages of the study could be appreciated better if the y-axis scale (ms) of all three graphs was the same. Also, I would suggest removing negative values (graph (a) starts at -10).
  • In the results section, the findings are described correctly, but often the reference of the Table(s) where the results are located is missing. Their addition would significantly improve the readability of the manuscript.

Author Response

Dear Reviewer,

I would like to thank you for your comments on the manuscript ”Right ventricular diastolic dysfunction after marathon run”. Please, find the answers below.

Point 1:

Is there a specific reason for including only male runners in the study? The fact that men and women have different left ventricular dimensions and function is well-established. Furthermore, sex differences associate with cardiac adaptations to endurance exercise. Thus, failing to include both sexes could be a limitation of the study and should be addressed as part of the discussion.

Response 1:

As you suggested, we included limitation of the study in lines 316-318: our study has an important limitation. As we did not include women in the study group, we cannot apply our findings to the population of female runners. Therefore, we are not able to discuss possible gender differences in RV diastolic fatigue caused by marathon run.

Point 2:

Figure 2. The changes in the heart rate-adjusted isovolumic relaxation time (IVRTc_RV, IVRTc_IVS, and IVRTc_LW) between the three stages of the study could be appreciated better if the y-axis scale (ms) of all three graphs was the same. Also, I would suggest removing negative values (graph (a) starts at -10).

Response 2:

As you recommended, we modified Figure 2 according to your suggestions. We corrected and unified the y-axis scale (ms) of all three graphs and we also removed negative values.

Point 3:

In the results section, the findings are described correctly, but often the reference of the Table(s) where the results are located is missing. Their addition would significantly improve the readability of the manuscript.

Response 4:

We also modified the results section following your suggestion and we added the references to the Table(s) where it was possible.

All changes to the manuscript text are tracked in the revised manuscript file with the Track Changes' mode.

We hope that you will find our paper interesting and you will approve it for publication.

Yours faithfully,

Assoc. Prof. Alicja Dąbrowska-Kugacka, MD PhD

Zuzanna Lewicka, MD